# Smooth Muscle Tumor of Uncertain Malignant Potential (STUMP): A Comprehensive Multidisciplinary Update

**DOI:** 10.3390/medicina59081371

**Published:** 2023-07-27

**Authors:** Andrea Tinelli, Ottavia D’Oria, Emanuela Civino, Andrea Morciano, Atif Ali Hashmi, Giorgio Maria Baldini, Radomir Stefanovic, Antonio Malvasi, Giovanni Pecorella

**Affiliations:** 1Department of Obstetrics and Gynecology and CERICSAL (CEntro di RIcerca Clinico SALentino), “Veris delli Ponti Hospital”, 73020 Scorrano, Italy; 2Department of Medical and Surgical Sciences and Translational Medicine, Sapienza University, 00185 Rome, Italy; ottaviadr@gmail.com; 3Department of Biological and Environmental Science and Technology, University of Salento, 73100 Lecce, Italy; emanuela.civino@unisalento.it; 4Department of Gynaecology and Obstetrics, Pia Fondazione “Card. G. Panico”, 73039 Tricase, Italy; drmorciano@gmail.com; 5Department of Histopathology, Liaquat National Hospital and Medical College, Karachi 74800, Pakistan; atifhashmi345@gmail.com; 6MOMO’ FertiLIFE, 76011 Bisceglie, Italy; gbaldini97@gmail.com; 7Department of Histopathology, University Clinical Centre of Serbia, 11000 Belgrade, Serbia; r_stefanovic@hotmail.com; 8Department of Biomedical Sciences and Human Oncology, University of Bari, 70121 Bari, Italy; antoniomalvasi@gmail.com; 9Department of Obstetrics, Gynecology and Reproductive Medicine, Saarland University, 66421 Homburg, Germany; giovannipecorella@hotmail.it

**Keywords:** smooth muscle tumors of uncertain malignant potential, STUMP, uterine fibroids, leiomyosarcoma, uterine mesenchymal tumors

## Abstract

*Background and Objectives*: The uterine smooth muscle tumors of uncertain malignant potential (STUMP) are tumors with pathological characteristics similar to leiomyosarcoma, but that do not satisfy histological criteria for leiomyoma. These are problematic lesions with intermediate morphologic features; thus, diagnosis and treatment are difficult. This narrative review aims to review data in the literature about STUMPs, particularly focusing on management and therapeutic options and strategies for women who desire to preserve fertility. *Material and Methods*: authors searched for “uterine smooth muscle tumor of uncertain malignant potential” in PubMed and Scopus databases, from 2000 to March 2023. Pertinent articles were obtained in full-text format and screened for additional references. Only articles in English language were included. Studies including full case description of patients with histopathological diagnosis of STUMP in accordance with Stanford criteria were included. *Results*: The median age was 43 years old. Symptoms are similar to those of leiomyomas, with a mean diameter of 8.0 cm. Total hysterectomy with or without bilateral salpingo-oophorectomy is the standard care for women if fertility desire is satisfied. Myomectomy alone can be considered for young patients. Although these tumors have not a high malignant potential, several studies described recurrence and metastases. *Conclusions*: STUMPs are complex uterine smooth muscle tumors, with a rare but reasoned clinical–diagnostic management. Considering the high clinical and histological complexity of these tumors, high level of expertise is mandatory.

## 1. Introduction

In 1973, Langley was the first to speak about “smooth muscle tumor of uncertain malignant potential”; he was the first to describe what we nowadays call STUMP [1]. The purpose of this manuscript is to give an identity to this not yet well-known pathological entity. We give an overview from many different points of view: the micro and microscopical specific features, the diagnostic instrumental findings, treatments, and follow-up. According to the World Health Organization [WHO] classification, uterine smooth muscle tumors of uncertain malignant potential [STUMP] are rare tumors with pathological characteristics easily confused with leiomyosarcoma (LMS), but that do not satisfy criteria for leiomyoma (LM), according to the 2014 WHO classification [2]. A STUMP can be defined as a uterine smooth muscle cancer that cannot be diagnosed unequivocally as benign or malignant. Typical histopathologic features were proposed from Stanford in 1994, including cytologic atypia, mitotic count, and tumor cell necrosis [3]. Although these tumors have not a high malignant potential, recurrence and metastases are described. The classification of uterine mesenchymal tumors remains a challenge, with considerable overlap regarding terminologies such as STUMP, atypical leiomyoma, atypical leiomyoma with low risk of recurrence 1.9% [4], and atypical leiomyoma with low malignant potential. This can lead to an equivocal diagnosis. A recent important contribution for the histological parameters of STUMP was provided by Gupta et al., reporting atypical mitoses, epithelioid differentiation, vascular involvement, and irregular margins [5,6]. Because they are problematic lesions with intermediate morphologic features, diagnosis and treatment is difficult. No accepted guidelines have been defined for diagnosis, treatment, and follow-up of the disease, and the management of uterine STUMP is considered a challenge. Total hysterectomy with or without bilateral salpingo-oophorectomy is the standard care for women if fertility desire is satisfied. Myomectomy alone can be considered for young patients [7,8]. The purpose of the study was to review all of the literature on STUMPs with the collaboration of several specialists, in order to have the most comprehensive picture possible under many aspects: biological, molecular, histological, immunohistochemical, gynecological, obstetric, etc.

## 2. Material and Methods

The present study aims to review data in the literature about STUMPs, particularly focused on management and therapeutic options and strategies for women who desire to preserve fertility. Authors searched for “uterine smooth muscle tumor of uncertain malignant potential” in PubMed and Scopus databases, from 2000 to March 2023. Pertinent articles were obtained in full-text format and screened for additional references. We led a first screening using, in the research database, the keyword “STUMP”; more than 20,000 abstracts appeared. That is why we decided to increase the specificity of our first screening using two keywords: “UTERINE STUMP”, reducing the abstract pool to 8000 manuscripts. The articles not in English language have been excluded. Some manuscripts show repetition or not useful information; we excluded them too. Studies including full case descriptions of patients with histopathological diagnosis of STUMP in accordance with the Stanford criteria were included. Finally, 52 manuscripts were included.

## 3. Anatomopathological Features

Macroscopically, STUMPs resemble fibroids and, thus, have a well-circumscribed and non-encapsulated appearance (Figure 1); however, sometimes the borders can be pushing or even focally permeative (Figure 2). The cut surface is usually tan-white, firm, and whorled. Areas of infarction, hemorrhage, and myxoid change can be apparent. The median size is 6.7 cm (range: 2.5–12.2 cm) [7]. Extensive sampling of these tumors is needed, specifically if myxoid areas are present. Microscopically, STUMPs are characterized by interlacing fascicles of spindle cells with cigar-shaped nuclei; however, the morphological features can vary widely (Figure 3).

STUMPs are basically tumors that fell short of unequivocal diagnosis of leiomyosarcoma, of which follow the FIGO classification (Table 1).

Table 1, the STUMPs have not their own FIGO classification, but they follow the one of the leiomyosarcomas.

Therefore, one of the four following criteria should be present for a diagnosis of STUMP [9,10,11]:smooth muscle tumors with focal/multifocal or diffuse cytological atypia (moderate-to-severe), lacking coagulative tumor necrosis and 6–9 mitoses per 10 HPFs (2–4 mitoses/mm^2^).Tumors that lack cytological atypia or raised mitotic count but having unequivocal coagulative tumor necrosis.Tumors with elevated mitotic count (>15 mitoses per 10 HPFs or >6 mitoses/mm^2^) but lacking coagulative tumor necrosis or cytological atypia.Tumors with uncertain mitotic count but having diffuse cytological atypia (moderate-to-severe).

The histological criteria for epithelioid and myxoid STUMPs vary considerably. Epithelioid smooth muscle tumors are characterized by round or polygonal cells with eosinophilic granular-to-clear cytoplasm. Epithelioid smooth muscle tumors with 2–3 mitoses/10 HPFs in the absence of coagulative tumor necrosis or atypia are classified as epithelioid STUMPs [12]. Alternatively, presence of one mitosis in the absence of coagulative tumor necrosis and atypia fulfills the criteria for myxoid STUMPs (Figure 4, Figure 5, Figure 6, Figure 7 and Figure 8). Myxoid STUMPs are generally hypocellular and cells are separated by myxoid matrix that is Alcian blue positive [13].

## 4. Molecular Biological Characteristics

On a molecular level, STUMPs show genomic heterogeneity ranging from rare chromosomal alterations to high chromosomal instability, including chromosomal gains and copy number gains [14] However, the chromosomal alterations are low, compared to leiomyosarcoma. In diagnostically challenging cases, molecular studies can be applied to differentiate between STUMPs and leiomyosarcoma. From 7% to 34% of all uterine smooth muscle tumors can be classified both as STUMPs and as leiomyosarcoma. According to the AIOM guidelines of 2019 (updated in October 2019), evaluation of the expression of the progesterone receptor (PgR), p53, and Ki67 could help pathologists in the differential diagnosis between leiomyosarcoma, leiomyoma and STUMPs. However, other markers were evaluated in other research studies, in combination with those aforementioned, for differential diagnosis, such as p16. P16 is a tumor suppressor gene that plays a crucial role in cell cycle regulation and most authors report that immunohistochemical p16 expression increases with tumor aggressiveness. The expression of p16, p53, and Ki67 proteins seems to be higher in leiomyosarcoma than in STUMPs [14,15]. The p16, p53, and KI67 proteins seem to be the most useful immunomarkers for identifying clinically aggressive smooth muscle tumors. A recent meta-analysis of Travaglino et al. categorized immunohistochemical patterns of gynecological STUMP as “abnormal” vs. “wild-type” for p53, “diffuse” vs. “focal/negative” for p16, and ≥ 10% vs. 10% for ki67, concluding that p53 and p16 might be useful in the risk assessment of STUMP. They cannot be used alone as prognostic markers [16]. Other studies focused on B-cell Lymphoma 2 (Bcl2) protein, which regulates the apoptosis and may initiate cell replication, making the cell independent of growth factors. The Bcl-2 was expressed more in leiomyosarcoma than in leiomyosarcoma and STUMPs. Still, others IHC markers have been described in the literature, like matrix metalloproteinase 2 (MMP 2), cellular retinol-binding protein-1 (CRBP 1), EGFR, and galectin-3, but more studies are needed to find a panel that can be useful in differentiating STUMP from leiomyoma and leiomyosarcoma [17,18]. A new molecular approach consists of Array-Comparative Genomic Hybridization Analysis, trying to find a specific genomic profile of smooth muscle lesions with uncertain histological characteristics. Tumors with a genomic index < 10 (low level of chromosomal rearrangements) are classified as STUMPs with good outcomes. On the contrary, a genomic index > 10 and a complex genomic profile represent STUMPs with poor outcome [12]. The STUMPs have some characteristics in common with inflammatory myofibroblastic tumors (IMTs). The IMT is a rare malignant tumor with lymphoplasmacytic inflammatory pattern and myofibroblastic cells; one case report described in the literature reported a mistake between IMT and STUMP [17]. Both of them are very rare neoplasms and few case reports are reported; for this reason, diagnostic criteria are not clearly defined.

## 5. Instrumental Features and Diagnosis

The preoperative diagnosis is quite difficult and only the histological examination is diriment. In more than half of the cases, the diagnosis is incidental and is often post-operative, because MRI and pelvic ultrasounds are methods not able to pre-operatively differentiate benign from malignant tumors. The median age of diagnosis is 43 years [19]. Gadducci et al. conducted a review, concluding that the median age of patients with STUMPs is 41–48 years, up to 75 years. Symptoms are similar to leiomyomas: pelvic pain, dysmenorrhea, abnormal uterine bleeding, menorrhagia and anemia, infertility, dysuria, and pelvic pressure. In total, 82.5% of women affected by STUMP presented with at least a large uterine mass ≥ 5.0 cm of diameter. The risk factors are poorly understood [2,20,21].

## 6. Ultrasound Diagnostics

There are not specific ultrasound characteristics for STUMP compared with leiomyoma or leiomyosarcoma. However, ultrasound features of malignancy like highly vascularized mass, irregular outline, irregular pattern due to necrotic areas, can be present that they are indistinguishable from leiomyosarcoma [22]. A recent retrospective study evaluated preoperative ultrasound assessment of fourteen women receiving a histopathological diagnosis of STUMP: a pathognomonic description has not been recognized [23]. The classification reported in the literature, which achieved a consensus opinion, to describe sonographic features of myometrium and uterine masses is the MUSA (Morphological Uterus Sonographic Assessment). The STUMP showed high doppler enhancement because of high peri- and intralesional vascularization (color score of 3 or 4 according to the MUSA criteria) [24].

## 7. Radiological Diagnostics

The use of ultrasound and diffusion-weighted (DW)-magnetic resonance imaging (MRI) to accurately diagnose STUMP is limited too. For accurate differentiation between leiomyoma and STUMP, contrast-enhanced (CE) magnetic resonance imaging (MRI) seems to be more specific compared with DW (0.96 versus 0.36, *p* < 0.05) [25]. The use of serum lactate dehydrogenase (LDH) matched with dynamic MRI can be useful as instrument to differentiate leiomyoma from leiomyosarcoma [26]. The usefulness of 18F-fluorodeoxyglucose [FDG] positron emission tomography [PET]—computed tomography [CT] scan is limited in differential diagnosis [27]. The STUMPs and leiomyosarcomas had a typical hollow ball sign at 18F-FDG uptake; this sign is absent in leiomyomas and it is not pathognomonic for STUMPs [28] The role of f 18fluorodeoxyglucose [FDG] positron emission tomography (PET) and computed tomography (CT) scans is unclear. Ho et al. proposed the specific sign on the FDG PET (“hollow ball”), representing a zone of coagulative necrosis typical [28,29].

## 8. STUMP Management

No practical guidelines are currently available for STUMP management, treatment, and follow-up. Surgery is the standard therapy [16]; however, the role of hormonal therapy or chemotherapy is still unclear [6,18], since there are limited data on whether STUMPs are hormone sensitive or not.

If the STUMP is diagnosed at preoperative histological examination (hysteroscopy or myomectomy specimen), hysterectomy is the gold standard treatment for women with children [17,26]. The fertility-sparing approach reserved to women wishing pregnancy needs an accurate follow-up protocol, with a multidisciplinary team. It can be a myomectomy followed by hysterectomy, after the completion of fertility desire.

Hence, it is essential to consider the patient’s age and the desire of future pregnancy. If a fertility program is completed, the gold standard is total hysterectomy with or without bilateral salpingo-oophorectomy in a biopsy-proven STUMP case. Different approaches for total hysterectomy can be considered: abdominal, vaginal, or mini-invasive approach [6,15,27,30].

The choice of the surgical technique depends primarily on the type of the lesion, the characteristics of the patient, fragility index of women, and the skills of the surgeon [31,32,33]. Some authors failed to detect a relationship between the risk of recurrence and the clinical features such as patient age, ethnicity, smoking habit, or type of surgery (hysterectomy versus myomectomy).

## 9. Surgical Management

The laparoscopic approach, when performed by expert operators, is considered the best choice. In young patients who desire to preserve childbearing, myomectomy alone should be considered, followed by hysterectomy after childbearing [15,27,28]. No comparative studies are currently available between laparoscopic or laparotomic myomectomy approach for women affected by STUMP. In the case of myomectomy, close surveillance should be mandatory. Studies about the role of morcellation for STUMPs are limited. In a retrospective chart review, Mowers et al. assessed the consequences of inadvertent morcellation of STUMPs, concluding that surgical re-exploration procedures after morcellation have a high likelihood of detecting peritoneal implants, which can be benign or malignant [34]. A recent manuscript in the literature, reviews the rate of recurrence after cytoreductive approach, with the open question if an organ sparring surgery is suitable to this type of histological entity. The study shows that the recurrence rate is so low that in the risk-benefit balance, a tumorectomy could be taken in consideration, especially in patients wishing pregnancy. The results about the organ sparring surgery were brilliant if analyzed with a fertility focus. Of 42 patients with unfinished reproductive plans, 22 pregnancies were recorded among 17 women (40.5%) [35]. Therefore, morcellation must be avoided to prevent the risk of diffuse peritoneal implants, which can be benign or malignant [15,36]. Nowadays, the in-bag power morcellation is a salient topic in case of uncertain tumor. Wright et al. compared the number of, post hysterectomy, Sarcomas before and after the FDA (Food and Drug Administration) advice to morcellation in endo-bag. Despite the fact that the use of containment bags increased by 3% every quarter, the number of sarcomas reduced from 0.17% to 0.12 [37].

## 10. STUMPs’ Follow Up and Recurrence

Annual surveillance with imaging (chest, abdomen and pelvic CT) and has been suggested after hysterectomy. Ip et al. suggested a follow-up visit every 6 months for the first 5 years followed by annual control for additional 5 years [8,28,38].

The risk of recurrence of these tumors exists, despite the low malignant potential [27]. Guntupalli et al. in a retrospective review assessed that recurrence-rate during the follow-up interval was 7.3% [19]. Deohar et al. had looked at 21 patients diagnosed with STUMP, of which one patient was noted to have metastatic liver disease, three years after the primary surgery [39].

A review of the literature by Vilos et al. reported that 71 patients with STUMP treated with myomectomy alone experienced no recurrent disease after 1 to 216 months, and that residual tumor was found in 2 (14.3%) of the 14 patients in whom initial myomectomy was followed by hysterectomy [7]. Ly et al. had similar result, with a recurrence rate of 12% [40].

## 11. STUMPs’ Metastases

Metastases of STUMP are rare [6]. A retrospective review reported cases of leiomyoma that had metastasized the lungs; the same study included a case of recurrence with lung metastasis that was reported 24 years after total hysterectomy for STUMP [41]. A recent multicenter retrospective cohort study concluded that morcellation/fragmentation is an independent risk factor of recurrence, with mitotic count too.

The patient’s disease-free survival is conditioned not by the surgical approach (laparotomic/laparoscopic), but only by unprotected morcellation that is just related to minimally invasive surgery. Furthermore, other specific characteristics are related to worse prognosis, identifying a category of patients who should undergo closer follow-up: low progesterone receptor (PR) (<83%), diffuse p16 expression, high proliferation activity [39]. The same finding was confirmed by a systematic review by Di Giuseppe et al., with a median follow-up of patients with recurrence of 40 months (range 2–288). The rate of recurrence is not homogeneous among studies because of the difficult histological diagnosis, the limited number of patients, and the different follow-up periods. The uterus is the most affected site of recurrence, followed by lung, bone, liver, and peritoneum [28,42,43]. In the literature, a recent case report described a tumor weighing 1570 g, which, after the anatomopathological examination, was identified as, one of the three reported in the literature, STUMP pulmonary metastasis [44].

## 12. Metastases Management

Standard guidelines for recurrence treatment are not available, but surgical treatment seems to be the best option. Data about the role of adjuvant therapy and/or radiotherapy are lacking: four cases of recurrence were described, and they were treated with surgery and chemotherapy (doxorubicin and cisplatin are the most commonly used agent). An alternative therapeutic option is represented by endocrine therapy based on progesterone, aromatase inhibitors, or gonadotropin-releasing hormone analogues [3,15].

## 13. STUMPs, Fertility and Pregnancy

After conservative surgery for STUMPs, few cases of successful pregnancy have been described. Campbell et al. conducted a review of the literature, including five case reports, concluding that good clinical outcomes for both mother and baby are possible after a fertility-sparing myomectomy and successful pregnancy, in a patient diagnosed with a STUMP [45]. In these women affected by STUMPs with fertility desire and treated conservatively, the risk of recurrence must be evaluated, considering the realistic possibility of pregnancy. However, the 5-year overall survival rate is 92–100% [34]. Peters et al. [46] reported that 5-year disease-free survival [DFS] and 5-year OS were 66% and 92%, respectively, among 15 patients with STUMP versus 28% and 40%, respectively, for 32 patients with leiomyosarcoma. A recent multicenter study evaluated the obstetrics features of patients diagnosed with STUMP: fifty-seven patients were included; ten of these had fertility desire and seven pregnancies were recorded, confirming the safety of the fertility-sparing approach. However, 14% of patients had recurrence during follow up, so a careful approach is mandatory [47]. In a retrospective review by Ha et al., after fertility-sparing surgery, pregnancy was successful in 80% of STUMP patients, although the number of patients was small (four out of five); in these cases, more attention should be paid to follow-up, balancing the risk (rate of recurrence 10.5%) [10].

A single-center retrospective experience demonstrated that two of six patients who underwent myomectomy for STUMP wished to retain their fertility and both of them had delivered full-term live babies by cesarean section, without complications [48].

Concerning in vitro fertilization, the literature showed an interesting case report on a Caucasian patient with endocrinal disorders and poor ovarian reserve (a-MH anti-Mullerian hormone was by 0.8 ng/mL, FSH Follicle-stimulation Hormone by 12.2 mU/mL, estradiol E2 42 pg/mL). Fertility of the partner was tested and detected in normal range. Beyond the hormonal examination the woman received a TUS (transvaginal ultrasound) which guided macroscopical anatomical control. The TUS did not show any pathological feature. After the conclusion of the preliminary examination, they started the IVF treatment, with cycles of COS (controlled ovarian stimulation) and ICSI (Intracytoplasmic Sperm injection). A mass in the uterine cervical area was discovered during the TUS’s interval sonographic check; this mass was unrelated to their pre-treatment. An MRI confirmed the cervical mass, which, after the laparoscopic removal, revealed itself as a STUMP. The mentioned study is the first manuscript correlating COS to STUMP [49].

The recent retrospective study of Zhang et al. tried to investigate treatment, management, and prognosis of patients with STUMP during a period of 13 years. Thirty-one patients were enrolled, and the most common symptom of clinical presentation was menstrual disorder. Most representative immunohistochemical markers in tumors were ER, PR, and p16. Two cases of relapse were described, within 36 months [42]. An interesting case report described a STUMP which has been diagnosticated in the pregnancy. The woman was in the 25th week of her pregnancy when the tumor was diagnosed. The obstetrical staff of the Aristoteles University of Thessaloniki decided for a conservative approach with tight control of the pregnant patient, who already at the diagnostic time showed an oligohydramnios. The follow-up showed an anhydramnios which led to a cesarean section in the 30th week of gestation. A neonate weighting 1350 g and a 9-kg tumor resulted from the surgery. The team achieved a conservative approach in avoiding the hysterectomy [50].

## 14. Conclusions

From what has been said, it is clear that STUMPs are complex uterine smooth muscle tumors, with a rare but reasoned clinical–diagnostic management. Moreover, the numbers of cases of assessed STUMPs reported are limited in the scientific panorama. There are no National Comprehensive Cancer Network Clinical Practice Guidelines established for STUMP.

If the diagnosis of STUMP is confirmed by histological examination, hysterectomy is the gold standard treatment for those women who have completed their childbearing planning. For women who want to preserve their fertility, a conservative approach with an accurate surveillance protocol is recommended. Considering the high clinical and histological complexity of these tumors, a high level of expertise in the field of soft tissue gynecological neoplasms is mandatory (including a gynecologic oncologist with high-level expertise in soft tissue sarcoma).

## Figures and Tables

**Figure 1 medicina-59-01371-f001:**
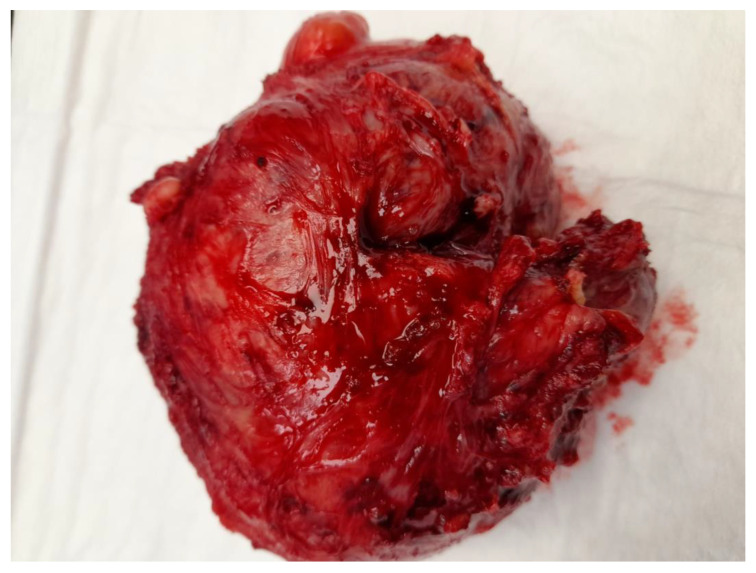
Image of a STUMP removed in laparotomy on a 39-year-old patient, having a well-circumscribed and non-encapsulated appearance.

**Figure 2 medicina-59-01371-f002:**
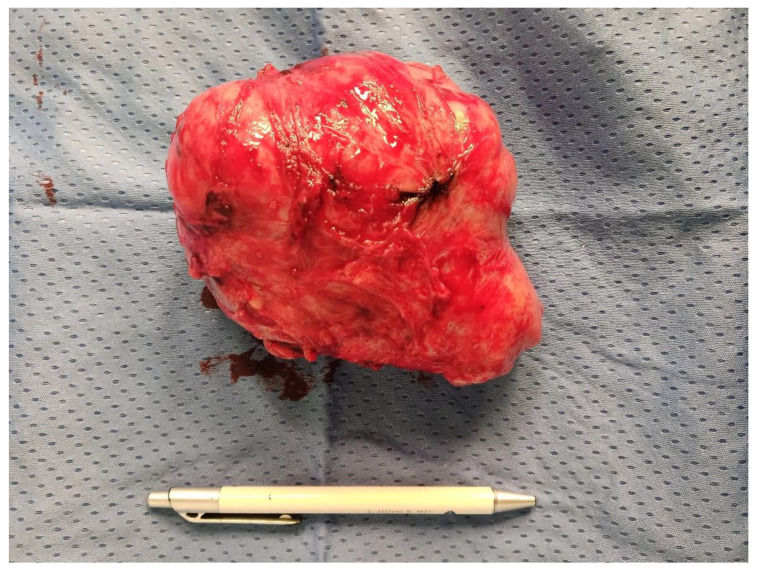
Image of another STUMP of consistent volume, removed in laparotomy in a young patient of 32 years; it appears soft in texture.

**Figure 3 medicina-59-01371-f003:**
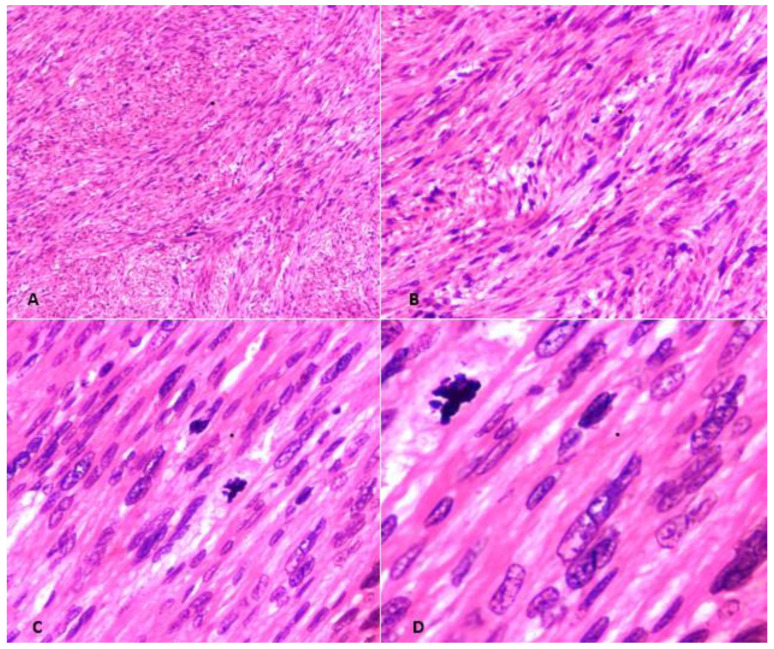
Smooth Muscle Tumor of Uncertain Malignant Potential (STUMP). (**A**) Hematoxylin and Eosin (H & E)-stained section at 40× magnification showing bundles of smooth muscle cells. (**B**) H & E-stained section at 100× showing tumor cells with diffuse moderate-to-severe atypia. (**C**) H & E section at 200× revealing spindled cells showing atypia and pleomorphism. (**D**) H & E section at 400× depicting atypia and occasional mitosis. There was no tumor necrosis and mitotic count was less than 10/10 HPFs, consistent with the diagnosis of STUMP.

**Figure 4 medicina-59-01371-f004:**
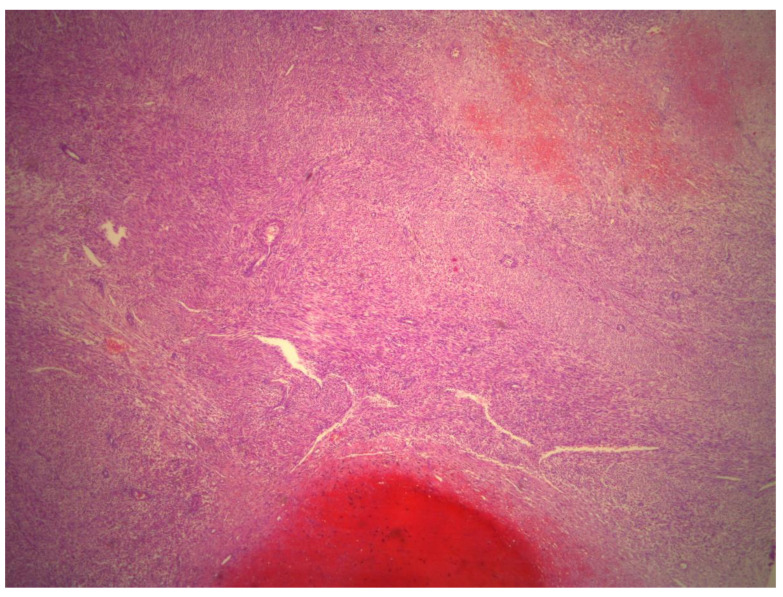
Histological magnification at 2.5×. The presence of necrosis that is difficult to classify: ischemic or coagulative.

**Figure 5 medicina-59-01371-f005:**
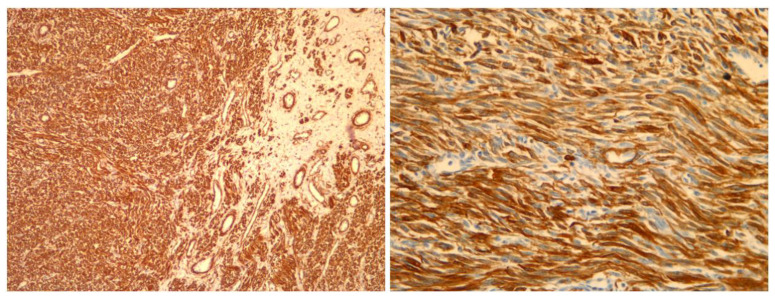
Image on the **left** (at magnification 10×) shows positive staining for SMA (smooth muscle actine); Image on the **right** (magnification 20×) shows positive staining for Caldesmon (immunoprofile characteristic for smooth muscle tumors).

**Figure 6 medicina-59-01371-f006:**
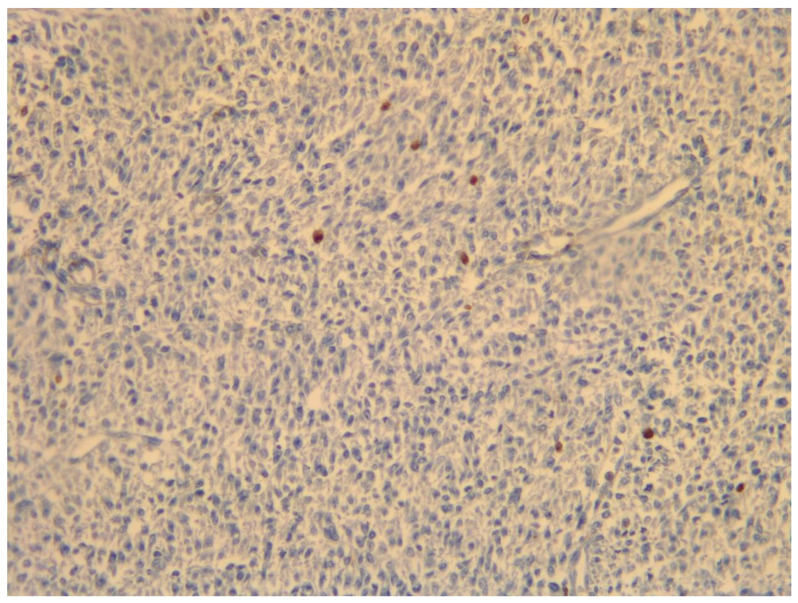
Histological magnification 20×: proliferative index (expression of ki-67) is variable and often not contributory; in this case, 1%.

**Figure 7 medicina-59-01371-f007:**
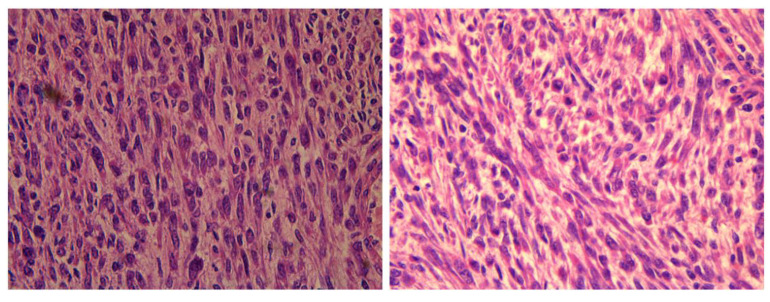
Image on the **left** (at magnification 40×) shows significant cytologic atypia. Image on the **right** shows low mitotic activity (in this particular case, 5–10 mitosis/10 HPF).

**Figure 8 medicina-59-01371-f008:**
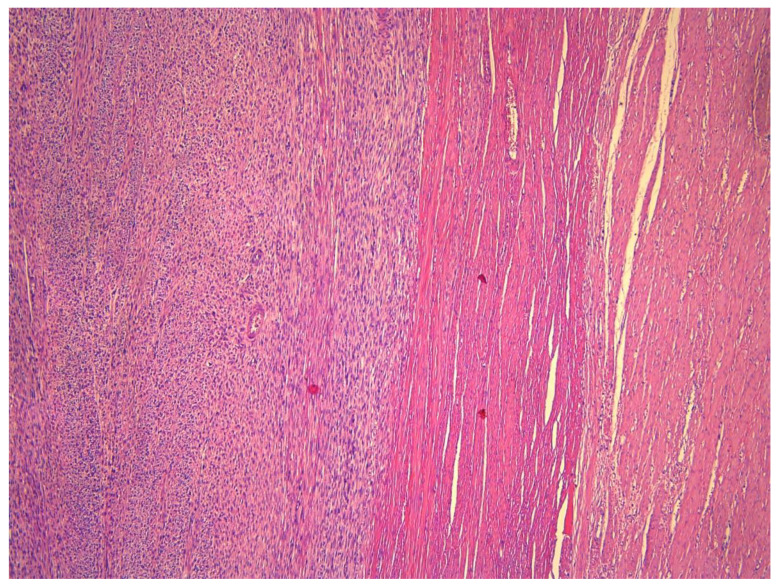
Histologic magnification 5×: This image shows sharp tumor demarcation (no infiltrative growth).

**Table 1 medicina-59-01371-t001:** FIGO classification.

FIGO I	Tumor limited in the Uterus
FIGO IA	Tumor ≤ 5 cm
FIGO IB	Tumor > 5 cm
FIGO II	Tumor limited in the Pelvis
FIGO IIA	involved adnexa
FIGO IIB	involved other extrauterine organ in the pelvis
FIGO III	Intraabdominal metastases
FIGO IIIA	one organ involved
FIGO IIIB	more than one organ involved
FIGO IIIC	positive pelvic/parietal Lymph nodes
FIGO IV	Involvement of bladder, rectum, extra pelvic metastases
FIGO IVA	Involvement of bladder, rectum
FIGO IVB	extra pelvic metastases

## Data Availability

No new data were created or analyzed in this study. Data sharing is not applicable to this article.

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
