# Peer review of "Smooth Muscle Tumor of Uncertain Malignant Potential (STUMP): A Comprehensive Multidisciplinary Update"

_medicina, 2023, doi:10.3390/medicina59081371_

Round 1

Reviewer 1 Report

The review article entitled 'Smooth Muscle Tumor of Uncertain Malignant Potential (Stump): A Comprehensive Multidisciplinary Up-To-Date' is a valuable study.

It is recommended to use 'Uncertain' instead of 'Undetermined' in the Figure 3 description.

I think that the information about the basic features of STUMP is given in the sub-title of 'Anatomopathological Features'. Sub-sections 3 and 4 can be rearranged to be more concise.

‘The expression of p16, p53 and Ki67 proteins seems to be higher in Leiomyosarcoma than in STUMPs and Leiomyosarcoma [15], and other authors found statistically significant differences in p16 expression between Leiomyosarcoma and TUMPs and between Leiomyosarcoma and Leiomyosarcoma, but no differences were observed between STUMPs and Leiomyosarcoma’ is a confusing sentence.

The sentence ‘Symptoms of STUMPs are similar to fibroid symptoms.’ can be removed because it is repeated later. (Line 184)

‘A case report described the attempt of treatment of an advanced cervical STUMP in a 58-years-old patient with an history of LASH (Laparoscopic supracervical hysterectomy). Due to the infiltration in the bladder and parametria, a curative surgical approach was unimaginable. The patient was treated with an adjuvant radio-chemotherapy in palliative approach (percutaneous radiation, as well as 6 cycles of Cisplatin 40mg/m² weekly, as radiosensitizer). An MRI post-treatment control showed a regression of the lesion, unfortunately a tumor recurrence showed itself in the CT scan after five months [30].’ I suggest you check and remove reference 30. (Line 226-232)

‘Lu et al compared 72 patients’ outcome of laparoscopy VS laparotomy for treatment of early-stage cervical stump carcinoma (Table 2). With respect to surgical complications, laparoscopy was associated with a significantly lower complication rate, less blood loss, a shorter operative time, and a higher hospitalization fee than laparotomy. Survival was not significantly different between the laparoscopy and laparotomy groups [39].’ (Line 270-274) Patients who had cervical cancer in the cervical stump after supracervical hysterectomy were included in this study. Please delete this reference and Table 2… This is not about smooth muscle tumor of uncertain malignant potential.

The phrase 'STUMP' should always be written in the same way in the text.

Although this review article aims to review the literature on STUMPs focusing on management and therapeutic options for women seeking to preserve fertility, it has been more of a book chapter. Emphasis was placed on immunological, molecular and histopathological findings rather than treatment strategies and follow-up. After all, it deviated from the purpose of this review article.

There are too many repetitive sentences in the article. please check them...

The quality of English Language is fine.

Author Response

The review article entitled 'Smooth Muscle Tumor of Uncertain Malignant Potential (Stump): A Comprehensive Multidisciplinary Up-To-Date' is a valuable study.

It is recommended to use 'Uncertain' instead of 'Undetermined' in the Figure 3 description.

Answer: according to reviewer suggestion the word “Undetermined” is replaced from “Uncertain” in the figure 3 description.

I think that the information about the basic features of STUMP is given in the sub-title of 'Anatomopathological Features'. Sub-sections 3 and 4 can be rearranged to be more concise.

Answer: We understand what suggested from the reviewer, but the authors estimated that it cannot summarize anymore the sub-sections. We used 27 lines to describe the micro and macroscopically features of STUMP and the cytological pathognomostic features. Regarding sub-section 4, we moved our focus on an even more microscopical level, describing the different protein, marker and the respective expression. Thus, the lines starting from 173, which as you underlined, expressed a confusing concept, have been compressed, to avoid misunderstandings.

‘The expression of p16, p53 and Ki67 proteins seems to be higher in Leiomyosarcoma than in STUMPs and Leiomyosarcoma [15], and other authors found statistically significant differences in p16 expression between Leiomyosarcoma and STUMPs and between Leiomyosarcoma and Leiomyosarcoma, but no differences were observed between STUMPs and Leiomyosarcoma’ is a confusing sentence.

Answer: according to reviewer suggestion, we rearranged the sentence in the following way: The expression of p16, p53 and Ki67 proteins seems to be higher in Leiomyosarcoma than in STUMPs and Leiomyosarcoma [15].

The sentence ‘Symptoms of STUMPs are similar to fibroid symptoms.’ can be removed because it is repeated later. (Line 184)

Answer: according to reviewer suggestion, it was removed.

‘A case report described the attempt of treatment of an advanced cervical STUMP in a 58-years-old patient with an history of LASH (Laparoscopic supracervical hysterectomy). Due to the infiltration in the bladder and parametria, a curative surgical approach was unimaginable. The patient was treated with an adjuvant radio-chemotherapy in palliative approach (percutaneous radiation, as well as 6 cycles of Cisplatin 40mg/m² weekly, as radiosensitizer). An MRI post-treatment control showed a regression of the lesion, unfortunately a tumor recurrence showed itself in the CT scan after five months [30].’ I suggest you check and remove reference 30. (Line 226-232)

Answer: We are not agreeing with the reviewer suggestion, since we should remove a reference of an innovative and recent manuscript. We underlined that we are speaking about a case report, of a experimental neoadjuvant chemotherapy. We are not proponing the one as treatment and to avoid misunderstanding, we didn’t deal the argument in the treatment section. If we didn’t understand your suggestion, we would please to give us further explanations.

‘Lu et al compared 72 patients’ outcome of laparoscopy VS laparotomy for treatment of early-stage cervical stump carcinoma (Table 2). With respect to surgical complications, laparoscopy was associated with a significantly lower complication rate, less blood loss, a shorter operative time, and a higher hospitalization fee than laparotomy. Survival was not significantly different between the laparoscopy and laparotomy groups [39].’ (Line 270-274) Patients who had cervical cancer in the cervical stump after supracervical hysterectomy were included in this study. Please delete this reference and Table 2… This is not about smooth muscle tumor of uncertain malignant potential.

Answer: The Table was updated but not removed. Many are in literature the manuscripts describing STUMP of the cervix, after LASH (1-3). It’s not the typical localization of this tumor of uncertain malignant potential, but the table is not focused on the localization of the STUMPs, otherwise on the surgical treatment independently on his localization.

The phrase 'STUMP' should always be written in the same way in the text.

Answer: according to reviewer suggestion, it was uniformed with STUMP acronym.

Although this review article aims to review the literature on STUMPs focusing on management and therapeutic options for women seeking to preserve fertility, it has been more of a book chapter. Emphasis was placed on immunological, molecular and histopathological findings rather than treatment strategies and follow-up. After all, it deviated from the purpose of this review article.

Answer: we agree with the reviewer, however we decided to perform this review analyzing, from all distinct professional corners, the recent (not extensive) STUMP literature, from all individual specialist's points of view. For this reason, the title of the paper is A COMPREHENSIVE MULTIDISCIPLINARY UP-TO DATE. We understand the reviewer's point of view, but our goal was different.

There are too many repetitive sentences in the article. please check them.

Answer: we tried to eliminate repetitions with the help of a native English speaker, who revised the text editing according to reviewer’ suggestion, for example a line number. We removed as requested from you the repetition of line 173.  

Reviewer 2 Report

Both in abstract and material and methods mention the number of cases included in the review. It will also be appropriate to mention number of cases excluded and the reasons for exclusion. Generally in a review clinical/specimen photographs are not included. We will have to mention source of these photographs and a brief clinical description of cases pertaining to these photographs. Acknowledgements for these photographs is must. Table-2 needs to be reorganized,age should come first, delete the line showing cost of surgery. A separate table showing in how many cases p51.p16 and other markers were done. Was there any death related to recurrence of tumor? list the investigations and procedure used for follow up. Need to revise text with help from English language person 

Needs to be checked by English language person

Author Response

Both in abstract and material and methods mention the number of cases included in the review. It will also be appropriate to mention number of cases excluded and the reasons for exclusion.

Answer: according to reviewer suggestion, we mention the number of abstracts include and excluded in the material and methods section.

Generally, in review clinical/specimen photographs are not included. We will have to mention source of these photographs and a brief clinical description of cases pertaining to these photographs. Acknowledgements for these photographs is must.

Answer: We didn’t mention or acknowledged any photograph, because all the images are part of the photographic repertoire of professor Tinelli (Author of the manuscript). According to reviewer suggestion we included this information and the Acknowledgements in the author contribution. Regarding the description of the clinical cases, we prefer not to mention them in the text, but leave only two images just to understand how the Stumps are macroscopically made.

Table-2 needs to be reorganized, age should come first, delete the line showing cost of surgery.

Answer: according to reviewer suggestion, we reorganized the table and deleted the surgery’s cost row.

A separate table showing in how many cases p51.p16 and other markers were done. Was there any death related to recurrence of tumor? list the investigations and procedure used for follow up.

Answer: We mentioned as you said the marked p16, otherwise we never spoke about p51 in our manuscript. Are maybe meaning p53? Regarding your request to list the investigations and procedure used for follow up, our 10th sub-section’s name is STUMPs’ Follow-up and recurrence, there we listed, the in literature suggested follow-up procedures. Regarding recurrence and dead, di Giuseppe et al. spoke about a 20% death ratio in case of STUMP recurrence (4).

Need to revise text with help from English language person.

Answer: native English speaker edited the text.

Reviewer 3 Report

This is a review article on smooth muscle tumor of uncertain malignant potential (STUMP).

The authors are to be commended for conducting a literature search and reviewing STUMP from a variety of perspectives.

The disease is a recently established concept, there are no standard guidelines worldwide, and clinicians face many difficulties in treatment and management.

This review discusses STUMP from various angles and provides one direction for clinicians dealing with this disease. The value of this paper is important because if many such reviews existed, they would contribute to the development of guidelines and other guidelines.

However, several similar reviews on STUMP have been reported. It would have been preferable to add to the initial introduction what makes this review different from others, what is the latest information, and why this report is being made. It would have added value to the paper to include this information.

There do not appear to be any major problems with the quality of the English in this paper.

Author Response

The authors are to be commended for conducting a literature search and reviewing STUMP from a variety of perspectives. The disease is a recently established concept, there are no standard guidelines worldwide, and clinicians face many difficulties in treatment and management. This review discusses STUMP from various angles and provides one direction for clinicians dealing with this disease. The value of this paper is important because if many such reviews existed, they would contribute to the development of guidelines and other guidelines. However, several similar reviews on STUMP have been reported. It would have been preferable to add to the initial introduction what makes this review different from others, what is the latest information, and why this report is being made. It would have added value to the paper to include this information.

Answer: we appreciate the real nice words used to describe our paper. We implemented in the sub-section introduction, as from reviewer suggested, in few words the goal of the manuscript, as the following: Purpose of this manuscript is to give an identity to this not yet well-known pathological entity. We gave a definition, the micro and microscopical specific features, the diagnostic-instrumental findings, treatment and follow-up.   

Round 2

Reviewer 1 Report

Dear Author, 

I've evaluated the revised version of the article titled 'Smooth Muscle Tumor of Uncertain Malignant Potential (Stump): A Comprehensive Multidisciplinary Up-To-Date'. I also carefully considered the authors' responses to my suggestions. In this review article, uterine smooth muscle tumors of uncertain malignant potential (STUMP) are discussed in all aspects.

The 'cervical stump', which is the subject of articles in references 30 and 39, represents the cervical tissue remaining after subtotal hysterectomy. Smooth muscle tumors of uncertain malignant potential, abbreviated as STUMP, and 'cervical stump' used in both articles do not have the same meaning. Cervical cancer cases (most of them are squamous cell carcinoma) were discussed in both articles. I think the subject will become clear when the full text of both articles is read.

In the revised text, the term 'stump' in references 30 and 39 has been changed to 'STUMP'. (Line 531, 563) I don't think this is true since that's not the original way the articles were citing.

 The quality of English language is average.

Author Response

In agreement with the reviewer, we have eliminated the references not relevant to the discussed topic; we realized what the reviewer rightly pointed out and immediately proceeded to eliminate the typo.